# *Bacillus subtilis*-Derived Postbiotics as a Multifunctional Bio-Catalyst for Enhancing Lactic Acid Bacteria Viability and Yogurt Quality

**DOI:** 10.3390/foods14101806

**Published:** 2025-05-19

**Authors:** Jing Wu, Zhilin Wang, Jingyi Hu, Jing Liu, Xueying Han, Hongping Chen, Siming Zhu, Junjin Deng

**Affiliations:** 1Xinjiang Key Laboratory of Biological Resources and Ecology of Pamirs Plateau, College of Life and Geographic Sciences, Kashi University, Kashi 844008, China; wujinghlh@sina.com; 2Agro-Biological Gene Research Center, State Key Laboratory of Swine and Poultry Breeding Industry, Guangdong Academy of Agricultural Sciences, Guangzhou 510640, China; wangzhilin@gdaas.cn (Z.W.);; 3School of Food Science and Engineering, South China University of Technology, Guangzhou 510640, China; 4Guangdong Laboratory for Lingnan Modern Agriculture Heyuan Sub-Center, Heyuan 517500, China

**Keywords:** lactic acid bacteria, *Bacillus subtilis*, postbiotics, viability, yogurt

## Abstract

This study demonstrates that *Bacillus subtilis* GDAAS-A32-derived postbiotics (BSP) enhance yogurt production by optimizing lactic acid bacteria (LAB) viability and functionality. BSP enhanced the growth kinetics and biomass accumulation of *Streptococcus thermophilus* and *Lactobacillus bulgaricus* in both an anaerobic and aerobic pure system. The addition of BSP significantly increased the viable cell counts of *S. thermophilus* and *L. bulgaricus*, milk-clotting activity, sensory properties, and extracellular polysaccharide content and improved the rheological properties. Moreover, BSP elevated viable counts of *S. thermophilus* and *L. bulgaricus* to 6.18 × 10^8^ CFU/g and 1.03 × 10^8^ CFU/g, respectively, by day 7—representing 11.3-fold and 9.3-fold increases versus controls at 20% supplementation. Metabolomic signatures confirmed peptidoglycan reinforcement and flavor enhancement. Mechanistically, BSP supplementation might reduce urate and H_2_O_2_ toxicity through NH_3_-mediated proton neutralization and oxygen scavenging, while establishing a pyrimidine salvage network and redirecting one-carbon metabolism, resulting in enhanced stress tolerance and significant improvements in bacterial viability.

## 1. Introduction

The global fermented dairy industry has witnessed exponential growth over the past decade, driven by increasing consumer awareness of functional foods and gut health benefits. Probiotics, defined as live microorganisms conferring health advantages when administered in adequate amounts, have become indispensable components in this sector. The global functional food market has witnessed exponential growth, with fermented dairy products accounting for 28% of total probiotic sales. Within this sector, yogurt stands as a technological and commercial benchmark, constituting a USD 59.7 billion industry [1] projected to expand at 4.8% CAGR through 2030. This commercial success stems from its unique combination of sensory appeal, nutritional value, and clinically validated health benefits—particularly in lactose digestion enhancement and gut microbiota modulation. In recent years, the development of functionally enhanced dairy products has emerged as a significant research focus within food science, particularly regarding the nutritional fortification of yogurt systems. This trend aligns with growing consumer demand for foods offering targeted health benefits beyond basic nutrition. Furthermore, existing studies have demonstrated promising approaches in functional yogurt modification [2,3]. While these investigations have successfully enhanced specific functional properties, there remains an opportunity to optimize synergistic combinations of bioactive components through probiotic viability enhancement.

The biochemical architecture of yogurt production hinges on a finely tuned symbiosis between two lactic acid bacteria (LAB): *Streptococcus thermophilus* and *Lactobacillus delbrueckii* subsp. *bulgaricus* [4]. These thermophilic LAB species exhibit a unique proto-cooperation mediated through metabolic cross-feeding. Phase I: *S. thermophilus* initiates rapid acidification via lactose hydrolysis, generating lactic acid and CO_2_ while secreting formate, folate, and pyruvate to stimulate *L. bulgaricus* proliferation [5,6]. *S. thermophilus* initiates fermentation by hydrolyzing lactose into glucose and galactose, generating lactic acid that lowers the pH to 4.5–4.6. This acidification triggers casein micelle aggregation, forming yogurt’s characteristic gel structure. Concurrently, *L. bulgaricus* contributes proteolytic activity through cell envelope proteinases, releasing essential amino acids (valine, histidine, and methionine) that compensate for *S. thermophilus*’s auxotrophic requirements. Recent metabolomic studies reveal that this symbiosis extends beyond nutrient exchange: *S. thermophilus* secretes formate and CO_2_ to stimulate *L. bulgaricus* growth, while the latter produces bacteriocins that suppress competing microbiota [7]. This intricate metabolic interplay directly determines yogurt’s organoleptic properties, including acidity (0.9–1.2% lactic acid), viscosity (>2000 mPa·s), and flavor compound profiles [5].

Despite these biological advantages, industrial yogurt production faces critical challenges in maintaining probiotic viability. Post-fermentation processes including centrifugation, homogenization, and cold chain logistics that expose LAB to oxidative stress (ROS accumulation) and mechanical shear forces, reducing viable counts by 1–2 log cycles [1]. Furthermore, only 20–40% of ingested probiotics survive gastrointestinal transit due to gastric acidity and bile salt emulsification [8,9]. Regulatory frameworks exacerbate these technical demands: China’s GB mandates ≥1 × 10^6^ CFU/g viable LAB throughout shelf life, while Japan’s JAS requires ≥1 × 10^7^ CFU/g at consumption. Conventional approaches to meet these standards—such as microencapsulation or cryoprotectant addition—increase production costs while potentially altering product texture. Thus, enhancing intrinsic LAB robustness through metabolic modulation represents a cost-effective alternative.

Beyond regulatory compliance and product functionality, maintaining LAB viability holds critical scientific and commercial significance. First, live LAB serve as active producers of postbiotic metabolites (e.g., short-chain fatty acids and bacteriocins) during storage and digestion, which directly modulate gut microbiota composition and immune responses. Clinical trials have demonstrated that ≥10^9^ CFU/dose of viable LAB is required to elicit measurable improvements in lactose intolerance and inflammatory bowel disease remission rates [10]. Second, viable cell counts directly influence consumer trust, as most of the functional food purchasers actively verify “live and active cultures” claims on labels. Third, strain-specific viability determines technological competitiveness, while a higher number of viable LAB has a higher fault tolerance in post-processing. These multifaceted demands underscore why LAB viability transcends mere regulatory compliance, impacting both therapeutic efficacy and market success.

Various methods have been developed to improve LAB activity, including the use of prebiotics and strict anaerobic conditions [11]. Emerging research highlights postbiotics—”preparations of inanimate microorganisms and/or their components that confer health benefits” [12]—as promising adjuvants for dairy fermentation. Unlike probiotics, postbiotic components including bacterial lysates, extracellular vesicles, and metabolic byproducts exhibit superior thermal stability and pH tolerance. Studies have highlighted that the short-chain fatty acid in post-biogenic components can provide extra energy to bacteria [13], while acetate can act as a growth agent [14], and β-glucan can enhance the effectiveness of probiotics [15]. These findings suggest postbiotics could potentiate LAB activity through both nutritional and epigenetic mechanisms.

Within this context, *Bacillus subtilis* emerges as an ideal postbiotic source. This generally regarded as safe (GRAS)-certified bacterium produces a repertoire of bioactive compounds [16]. From 1999 to 2018, numerous *Bacillus* strains, including postbiotics from their inactivated and heat-killed cell preparations, were listed as GRAS, with 67 entries. The National Food Safety Standard of China, GB 2760-2014, also explicitly includes a variety of enzymes sourced from *B. subtilis* among its approved list of food additives [17]. *Bacillus* are known to produce bioactive molecules such as hydrolytic enzymes, antioxidant enzymes, and surface proteins, all of which are beneficial for the growth and survival of LAB. Recent studies have elucidated the crucial role of *B. subtilis* in supporting the heme-dependent catalase for the remission of the damage of reactive oxygen species against *Lactobacillus* [18]. Notably, research indicates that *B. subtilis* may activate heme-dependent catalase, mitigating reactive oxygen species damage to LAB, thereby exerting probiotic effects [16,19,20].

Previously, we isolated a *B. subtilis* strain, GDAAS-A32, from poultry intestine [21], which demonstrated a strong capability to promote the growth of various LAB under both aerobic and anaerobic conditions in vitro and in vivo. The intestinal origin of GDAAS-A32 confers ecological compatibility with LAB consortia, suggesting evolutionary adaptation to microbial cross-interaction mechanisms within nutrient-rich gastrointestinal niches—a trait hypothesized to enhance its functional synergy with dairy fermentation systems. In this study, the potential of postbiotics derived from *B. subtilis* GDAAS-A32 (BSP) to enhance the growth and metabolic activity of LAB during yogurt fermentation was explored. The sensory evaluation, acidification activity, volatile compounds, and microbial activity in yogurt fermented with or without BSP addition were compared. This comprehensive analysis provides valuable insights into the physiological and interactive properties of this novel postbiotic, guiding its potential application in large-scale industrial yogurt production.

## 2. Materials and Methods

### 2.1. Microorganisms

*S. thermophilus* GDMCC 1.2800 and *L. bulgaricus* GDMCC 1.1801 was obtained from Guangdong Microbial Culture Collection Center. *B. subtilis* GDAAS-A32 was maintained in our laboratory. Long-term conservation was achieved through preparation of master cell banks using a cryoprotective medium consisting of 15% (*v*/*v*) glycerol, with aliquots stored at −80 °C in temperature-monitored ultra-low freezers.

### 2.2. Preparation of BSP

To minimize matrix interference from complex nutrient sources, BSP production was conducted using chemically defined M9 medium. The process commenced with revitalization of *B. subtilis* GDAAS-A32 from cryopreservation (−80 °C) through streaking on LB agar plates. After incubation at 37 °C for 12 h, a single colony was selected and transferred to M9 liquid medium. The culture was incubated at 37 °C and 200 rpm until it reached an OD_600_ of 0.5–0.7. The metabolite-rich supernatant was subsequently separated through differential centrifugation (6000× *g*, 10 min, 4 °C). To comply with the ISAPP criteria defining non-viable microbial preparations, the supernatant was sterilized through 0.22 μm polyethersulfone membrane filtration to generate the BSP. Sterility verification was rigorously performed by inoculating LB with filtered BSP and monitoring microbial growth through 72 h turbidimetric analysis at 37 °C with continuous agitation (150 rpm), confirming complete absence of culturable microorganisms.

### 2.3. Effect of BPS on the Growth of S. thermophilus and L. bulgaricus

Preliminary assessments indicated not particularly obvious growth-promoting effects of BSP on LAB in nutrient-replete MRS medium. This attenuated response was attributed to nutritional saturation inherent to the MRS formulation, which likely obscured potential stimulatory interactions through comprehensive metabolic provisioning. To isolate BSP-mediated effects, we transitioned to a modified MC medium (5 g/L tryptone, 3 g/L beef extract, 3 g/L yeast extract, and 20 g/L glucose) that supports controlled microbial proliferation. This formulation provides suboptimal nutrient availability while maintaining biomass yield stability, thereby enabling detection of growth modulation. Individual cultures of *S. thermophilus* and *L. bulgaricus* were activated in appropriate growth medium until they reached an optical density at 600 nm of approximately 1.0. A 3% inoculum of each activated culture was transferred to modified MC medium. BSP was then added to the inoculated medium to a final concentration of 20% (*v*/*v*). The inoculated cultures were incubated at 37 °C, and OD_600_ measurements were taken every 2 h for a total duration of 24 h. Growth curves were generated by plotting OD_600_ values against time.

### 2.4. Yogurts Preparation

The process of yogurt preparation is shown in Appendix A. Skim milk powder was reconstituted with water (20% *w*/*v*), pasteurized at 65 °C for 30 min, and cooled to room temperature. The milk was then supplemented with 10% or 20% BSP and refrigerated at 4 °C. Single colonies of *S. thermophilus* and *L. bulgaricus* were separately cultured in M17 or MRS broth at 37 °C until OD_600_ of 1.0 was reached. The two cultures were mixed at a 20:1 ratio (*S. thermophilus*: *L. bulgaricus*) and inoculated into the prepared milk [22]. Fermentation was carried out at 42 °C for 6–8 h, or until the pH reached 4.60.

### 2.5. Enumeration of Biomass and Cell Counts

Viable cell counts were determined using the standard plate count method. *S. thermophilus* and *L. bulgaricus* were selectively enumerated using M17 and MRS medium at 37 °C for 24 h aerobically, respectively. The biomass of *S. thermophilus* and *L. bulgaricus* was quantified using qPCR. Primers specific for each species [23,24] were used to amplify target genes, and a standard curve was generated using a dilution series of known cell concentrations. The biomass of the samples was then calculated based on the Ct values obtained from qPCR and the standard curve. The DNA samples were extracted from bacterial cultures of LAB with a TIANamp Bacteria DNA kit (Omega Bio-tek Inc., Norcross, GA, USA) according to the manufacturer’s instructions. The PCR reaction components included 12.5 μL of TB Green (Takara, Beijing, China), 0.5 μL of each primer (10 μmol/L), 0.4 μL of DNA template, and 11.1 μL of RNase free ddH_2_O. The PCR reaction procedure was as follows: 95 °C for 1 min, followed by 30 cycles of 95 °C for 30 s, 60 °C for 1 min, 60 °C for 30 s, and finally 90 °C for 1 min.

### 2.6. Sensory Evaluation of Yogurts

Prior to conducting the evaluation, training is required for selecting members in four steps: 1. selection of sensory acuity to test basic perception; 2. theoretical training to master evaluation standards, terminology, and methods (e.g., the three-point test method); 3. systematic training of sensory thresholds (taste, smell, touch, etc.) and calibration of sensitivity through blind testing; 4. practical assessment to simulate real-life scenarios for evaluating the product and establishing scale consistency. Regular retraining and statistical verification of evaluation reliability were used to eliminate subjective bias and ensure objectivity of results.

Ten trained panelists (5 men and 5 women) were selected to evaluate the yogurt. An overall preference test was conducted using a 9-point preference scale, encompassing the following descriptors: extremely unflavored, very unflavored, unflavored, less unflavored, neutral, favored, very favored, and extremely favored, corresponding to scores from 1 to 9 [25].

For detailed sensory profiling, ten trained panelists (5 men and 5 women) were selected. The sensory attributes—odor, flavor, texture, and taste—were evaluated on a 100-point scale. Panelists were instructed to cleanse their palates with water between samples to maintain consistency in scoring, following specific guideline [26].

### 2.7. Milk-Clotting Enzyme Activity

Milk-clotting activity was determined following the previous method, expressed in Soxhlet units (SU) [27]. In brief, 0.5 mL of sample was added to a test tube containing 5 mL of reconstituted skim milk solution (10 g dry skim milk/100 mL, 0.01 M CaCl_2_) pre-incubated at 35 °C for 5 min. The solution was mixed well, and the clotting time T (s), measured as the time period from the addition of test material to the first appearance of clots, was recorded, and the clotting activity was calculated using the following formula:Soxhlet units (SU) = 2400 × 5 × D/T × 0.5(1)
where T = clotting time (s); D = dilution of test material. One SU is defined as the amount of enzyme required to coagulate 1 mL of a solution containing 0.1 g skim milk powder within 40 min at 35 °C.

### 2.8. Physicochemical Analysis of Yogurt

Viscosity was measured using an NDJ-5s viscometer.

Water-holding capacity (WHC) was assessed using a centrifugation method [28]. Briefly, 5.0 g of yogurt sample was weighed in a Falcon tube (Mi) and centrifuged at 3556× *g* for 30 min at 10 °C. After discarding the supernatant, the resulting precipitate was weighed (Mp). WHC was calculated using the following formula:WHC (%) = (1 − Mp/Mi) × 100(2)

The pH of yogurt samples was measured using a pH meter (Thermo Orion Model-420A). Acidity was determined according to the National Standards of the People’s Republic of China: Determination of Acidity in Food [29].

### 2.9. Quantification of Extracellular Polysaccharides (EPS) in Yogurt

EPS concentrations were quantified using the phenol-sulfuric acid method [30]. Briefly, 10 g of yogurt sample was weighed, 4000× *g*, and centrifuged for 15 min. The supernatant was collected and was adjusted to pH 7.5. Subsequently, 1/10 volume of trypsin solution (3 g/L) was added, and the mixture was incubated at 40 °C for 2.5 h to remove protein. Following this, 1/3 volume of Sevag solution was added, and the mixture was vigorously oscillated for 30 min and then centrifuged for 30 min. The supernatant was collected, and EPS was precipitated by adding 3–5 volumes of cold ethanol. The precipitated EPS was washed 2–3 times with cold acetone to remove any residual contaminants. The purified EPS was then dissolved in an appropriate volume of sterile water. The concentration of EPS was determined by the phenol-sulfuric acid method using glucose as the standard. A standard curve was prepared with known concentrations of bovine serum albumin, and the absorbance of the EPS samples was measured at 490 nm. The EPS concentration in the yogurt samples was then calculated based on the standard curve.

### 2.10. Metabolomics Analysis of Yogurt

The yogurt samples were vortexed for 1 min and mixed evenly. The subsequent steps are detailed as follows: accurately transfer an appropriate amount of sample into a 2 mL centrifuge tube, add 200 µL methanol and 200 µL Methyl tert-butyl ether solution, and vortex for 1 min. Centrifuge for 15 min at 12,000 rpm and 4 °C, filter the supernatant with a 0.22 μm membrane, and transfer the supernatant into the detection bottle for LC-MS detection. The LC analysis was performed on a Vanquish UHPLC System (Thermo Fisher Scientific, Waltham, MA, USA). Chromatography was carried out with an ACQUITY UPLC ^®^ HSS T3 (2.1 × 100 mm, 1.8 µm) (Waters, Milford, MA, USA). The column was maintained at 40 °C. The flow rate and injection volume were set at 0.3 mL/min and 2 μL, respectively. Mass spectrometric detection was performed on an Orbitrap Exploris 120 (Thermo Fisher Scientific) with an ESI ion source. Simultaneous MS1 and MS/MS (Full MS-ddMS2 mode, data-dependent MS/MS) acquisition was used. The *m*/*z* range of the detected metabolites is between 100 and 1000.

### 2.11. Statistical Analysis

All experiments were conducted in triplicate or more, with each experiment repeated three times. Statistical analysis was conducted using ANOVA tests using IBM SPSS 22.0 Statistics software. A *p*-value < 0.05 was considered as statistically significant.

## 3. Results

### 3.1. BSP Enhances the Microbial Activity of S. thermophilus and L. bulgaricus

Microbial activity plays a crucial role in the quality and safety of fermented foods. Firstly, the impact of BSP on the growth of *S. thermophilus* and *L. bulgaricus*, key microorganisms in yogurt fermentation, was determined. The growth curve of starter cultures revealed enhancement by BSP supplementation. Under aerobic cultivation (Figure 1A), *S. thermophilus* with BSP exhibited accelerated logarithmic phase initiation, achieving a higher OD_600_ value at 10 h compared to the control. The stationary phase (>1.2) was reached 2 h later in BSP-treated groups (18 h vs. 16 h control) with a 1.66-fold higher OD_600_ plateau (0.98 ± 0.001 vs. 0.59 ± 0.008, *p* < 0.001). Similar effects were observed under anaerobic conditions, *S. thermophilus* supplemented with BSP displayed a significantly higher OD_600_ value at 4 h compared to the control (Figure 1B), maintaining a stable and superior growth rate, and reaching a plateau at 18 h with a maximum OD_600_ of 1.31.

For *L. bulgaricus*, aerobic growth with BSP (Figure 1C) showed shortened log phase entry and 21.86% higher final biomass (OD_600_ = 0.44 ± 0.003 vs. 0.36 ± 0.009 control, *p* < 0.01). Anaerobic cultures (Figure 1D) demonstrated rapid proliferation after 6 h, with 20% of the BSP group achieving OD_600_ = 0.51 ± 0.007 at 24 h (vs. 0.32 ± 0.025 control), maintaining a higher growth rate throughout fermentation.

### 3.2. BSP Enhances the Microbial Activity, Acidification Rate, and Quality in Yogurt Preparation

Microbial activity significantly influences the fermentation characteristics and post-acidification of yogurt. To assess the impact of BSP on microbial growth in yogurt preparation, the biomass of *S. thermophilus* and *L. bulgaricus* during fermentation was monitored using qPCR. For *S. thermophilus*, 20% BSP supplementation induced early exponential growth, with the sustained growth rate higher than control (Figure 2A). At 4 and 6 h, the biomass of *S. thermophilus* continued to increase, with the 20% BSP group maintaining a stable growth rate consistently and showing a significant advantage over the control. When the control group rapidly grew to 4 h, its biomass had already caught up with the 10% BSP group, and its growth rate rapidly decreased to the same level as the 10% BSP group thereafter. Similarly, the biomass of *L. bulgaricus* was significantly higher in the BSP-supplemented samples at 2 h (Figure 2B). At 4 h, the 20% BSP group showed a significant increase, while the 10% BSP group did not differ significantly from the control. At 6 h, the biomass of *L. bulgaricus* decreased in all samples, but the BSP-supplemented groups maintained higher levels than control.

By the end of fermentation, the 20% BSP group exhibited a significant increase in *S. thermophilus* of 6.71 × 10^8^ CFU/g and in *L. bulgaricus* of 2.56 × 10^8^ CFU/g compared to the control group. These results demonstrate that BSP significantly enhances the activity of both *S. thermophilus* and *L. bulgaricus* during yogurt fermentation. Acidification, a critical parameter in yogurt fermentation, is primarily driven by LAB. As shown in Figure 2C, the pH decreased steadily during fermentation. The 20% BSP group exhibited a significantly lower pH and a faster rate of acidification compared to the control group. The 10% BSP group showed no significant difference in pH compared to the control. These results suggest that BSP enhances the growth and metabolic activity of these bacteria, contributing to the accelerated acidification and improved overall quality of the yogurt.

The 20% BSP group displayed a milky white color, uniform texture, and no whey separation. The flavor was described as slightly sour, delicate, and rich in a fermented milk aroma, without any off-flavors. The 10% BSP group exhibited similar improvements, with a whiter color, more uniform texture, and minimal whey separation. In contrast, the control group had a yellowish color, a less uniform texture, and slight whey separation. The flavor was described as slightly sour and delicate, with a milder fermented milk aroma (Figure 2D). The enjoyment of yogurt is governed by personal perception of aroma, taste, and texture. Among these, flavor has the greatest effect on consumer acceptance and preference. Sensory evaluation revealed that BSP-supplemented yogurt had a superior appearance (Figure 2E).

Consumer preference ratings further supported these findings (Figure 2F). The 10% and 20% BSP groups received significantly higher preference scores compared to the control group, indicating that consumers preferred the yogurt with added BSP. While there was no significant difference in preference between the 10% and 20% BSP groups, both groups were significantly preferred over the control. These results highlight the potential of BSP to improve the overall quality and consumer acceptability of yogurt.

### 3.3. BSP Improves the Physicochemical Characteristic of Yogurt

Milk coagulation is a critical step in fermented dairy product production and involves the action of rennet, which cleaves the N-terminal region of κ-casein. This cleavage destabilizes casein micelles, leading to their aggregation and the formation of a coagulated gel. During yogurt production, we observed that the supplement of BSP accelerated the coagulation process 0.5–1 h in yogurt fermentation. This suggests that BSP may enhance the activity of milk-clotting enzymes. Thus, the milk-clotting enzyme activity in the yogurt samples was measured. The milk-clotting enzyme activity in the control group was 142.51 ± 3.96 SU/mL, while the 10% and 20% BSP groups exhibited activities of 178.48 ± 7.98 SU/mL (25.24% increase) and 267.27 ± 5.00 SU/mL (87.54% increase), respectively (Figure 3A). These results indicate that BSP can significantly enhance the activity of milk-clotting enzymes, leading to faster coagulation and improved yogurt production. 

The coagulum stability is an important quality parameter of yogurt. The supplement of BSP significantly improved both the viscosity and water-holding capacity of the samples (Figure 3B,C). The 20% BSP group exhibited the highest viscosity of 60,487 ± 456.88 mPa·s and water-holding capacity of 46.71 ± 0.06%, followed by 57,702 ± 1530.98 mPa·s and 45.73 ± 0.04% in 10% BSP group, both higher than 54,826 ± 929.44 mPa·s and 40.36 ± 0.02% in the control group. *Lactobacillus* EPS is a kind of natural polymer produced during the growth and metabolism of lactobacilli and secreted outside the cell. The yogurt with 20% BSP addition had the highest EPS content of 90.78 ± 2.61 mg/L (Figure 3D), while the 10% BSP (83.14 ± 1.56 mg/L) and control group (83.72 ± 1.06 mg/L) showed no significant difference. These findings demonstrate that the addition of BSP can significantly improve the texture and stability of yogurt.

The addition of BSP also positively impacted yogurt quality, as assessed by viable cell counts, appearance, and flavor. Consistent with the biomass measurements, the 10% and 20% BSP groups exhibited significantly higher viable cell counts, reaching 6.53 × 10^8^ and 2.28 × 10^9^ CFU/g for *S. thermophilus*, 4.44 × 10^8^ and 6.27 × 10^8^ CFU/g for *L. bulgaricus*, respectively, higher than the 2.93 × 10^8^ and 3.23 × 10^8^ CFU/g of the control (Figure 3E,F).

### 3.4. BSP Enhances Yogurt Profile

Metabolomic analysis was employed to differentiate between control yogurt and yogurt supplemented with 20% BSP. Principal component analysis (PCA) revealed distinct clustering, indicating significant differences in the metabolites between the two groups (Figure 4A). A total of 451 volatile flavor compounds out of a total of 1489 metabolites including alcohols, aldehydes, ketones, carbonyl compounds, organic acids, and other compounds were identified in the yogurt (Figure 4B). A comparative analysis of the two groups identified 400 differential metabolites, with 204 significantly upregulated and 196 significantly downregulated in the BSP-supplemented yogurt (Figure 4C). Notably, several flavor-related compounds, including four carboxylic acids, four alcohols, six esters, and six fatty acids were significantly upregulated (Figure 4D).

To elucidate the metabolic pathways associated with the identified differential metabolites, a pathway enrichment analysis was conducted using the KEGG database. A total of 27 pathways were significantly enriched, with ten of the most prominent pathways including toluene degradation, phenylalanine metabolism, GABAergic synapse, central carbon metabolism in cancer, D-amino acid metabolism, protein digestion and absorption, arginine biosynthesis, glucosinolate biosynthesis, and clavulanic acid biosynthesis (Figure 4E). Among these pathways, enhanced reactions were arginine decarboxylation, hypoxanthine oxidation, carbamoyl-phosphate synthesis (glutamine-hydrolyzing), thymidine phosphorolysis, tyrosine decarboxylation, trans-cinnamate oxidation, and muconate cycloisomerization. The weakened reactions were xanthine oxidation, clavaminic acid synthesis, formiminoglutamate deamination, phloroglucinol reduction, pteridine reduction, *N*-isopropylammelide hydrolysis, and *N*-isopropylammelide reduction.

### 3.5. BSP Delays the Acidification of Yogurt During Storage

The experimental protocol excluded extended 7-day refrigeration trials to mitigate potential spoilage risks arising from cumulative temperature fluctuations and sampling-induced environmental perturbations that could compromise microbial stability parameters. The pH continued to decrease during storage (Figure 5A). While no significant difference was observed between the control group and the 10% BSP group (*p* > 0.05), the 20% BSP group exhibited a significantly slower pH decrease rate (*p* < 0.05). Although the 20% BSP group initially had a lower pH, its pH was higher than the other two groups after 7 days of refrigeration, indicating that its rancidity was inhibited. The changes in TA further illustrate this trend during storage (Figure 5B). The TA of all yogurt samples increased over time, but the 20% BSP group maintained the lowest TA (55°T) at the end of storage, followed by the 10% BSP group (58.5°T) and the control group (63.25°T). This suggests that BSP supplementation can slow down the rate of yogurt souring.

The viable cell counts of *S. thermophilus* and *L. bulgaricus* during storage are depicted (Figure 5C,D). Both species showed a gradual decline in viable cell counts over time, but the BSP-supplemented groups maintained higher levels compared to the control group. Notably, the 20% BSP group exhibited significantly higher counts of both bacteria, with final counts exceeding 6.18 × 10^8^ CFU/g for *S. thermophilus* and 1.03 × 10^8^ CFU/g for *L. bulgaricus* on day 7, higher than the 5.49 × 10^7^ and 1.11 × 10^7^ in the control group. In addition, BSP promoted the survival ability of LAB after refrigerated storage for 7 days. The survival rates of *S. thermophilus* and *L. bulgaricus* in the 20% BSP group were 26.9% and 53.6%, while their survival rates were 38.1% and 24.6% in the 10% BSP group, significantly higher than the 18.6% and 11.3% in the control. The fortified yogurt sample was just slightly decreased without significant differences (*p* > 0.05) after refrigerated storage for 7 days and was significantly higher than those in the control (*p* < 0.05). This indicates that BSP can prolong the shelf life of yogurt by maintaining a higher viable cell count. The recurrent sampling procedures implemented during the study period introduced a quantifiable risk of extrinsic microbial contamination, as evidenced by irreversible physicochemical deterioration, which precluded further analysis beyond day 7 to maintain data accuracy. As a precautionary measure, all experimental endpoints were systematically terminated at the 7-day mark.

## 4. Discussion

Fermented dairy products, such as yogurt, are complex microbial ecosystems that rely on the symbiotic relationship between LAB for successful production [31]. Yogurt, in particular, is typically fermented by *S. thermophilus* and *L. bulgaricus*. According to the Codex Standard for Fermented Milks [32], yogurt must contain viable and active cultures of these bacteria throughout its shelf life. The microbial activity of these bacteria significantly influences the fermentation properties and quality of yogurt. Enhancing the growth and metabolic activity of these strains is crucial for achieving desired yogurt characteristics [33]. However, strategies to enhance microbial activity in yogurt production have been limited. At present, common methods of increasing the viable count of *Lactobacillus* usually involve adding growth-promoting substances such as oligosaccharide [34,35], yeast mannans [36], yeast extract [37], storing the product in a glass container [38], and increasing the concentration of the starter [37,39]. The suitable strains were selected on the basis of acid- and bile-resistance [40], two-step fermentation [41], and so on. Furthermore, to enhance their viability during food storage and digestion, strategies such as pre-exposure or pre-cultivation of probiotic microorganisms under stressful conditions, including acidic, osmotic, and oxidative stress, have been implemented [42,43]. However, these methods will increase the cost of yogurt, and their effects to improve the vitality of probiotics were limited.

In a previous work, *B. subtilis* GDAAS-A32 was isolated from poultry intestinal microbiota, demonstrating exceptional probiotic potential through its capacity to enhance LAB proliferation [21]. While chromosomal architectures showed conserved synteny (99.98% similarity), GDAAS-A32 harbored an endogenous plasmid absent in the reference strain 168. Comprehensive biosecurity assessment via pathogenicity island prediction (PIPS), virulence factor database (VFDB), and comprehensive antibiotic resistance database (CARD) screening confirmed the absence of virulence factors and antibiotic resistance genes in the plasmid. In this study, the introduction of GDAAS-A32-derived postbiotics (BSP) fundamentally altered the microbial ecology of yogurt fermentation systems. Plate enumeration revealed 11.26- and 9.28-fold increases in *S. thermophilus* and *L. bulgaricus* viability with 20% BSP supplementation. This growth potentiation correlates with the metabolomic findings showing a 92-fold elevation in *N*-acetylmuramic acid, a peptidoglycan precursor critical for cell wall biosynthesis.

While conventional yogurt production relies on controlled bacterial growth to maintain optimal texture and physicochemical properties, our systematic quality assessment addressed potential concerns regarding BSP-induced hyperproliferation. Excessive microbial activity could theoretically compromise product integrity through acid overproduction, protein matrix destabilization, and impaired water retention capacity. Comprehensive analysis of fermentation kinetics and storage stability revealed that BSP supplementation paradoxically enhanced key functional characteristics despite elevated viable counts. The enhanced metabolic activity manifested through two primary pathways: proteolytic processing of milk proteins and exopolysaccharide (EPS) biosynthesis. BSP-enriched fermentations exhibited 87.54% greater milk-clotting activity (Figure 3A), correlating with significantly elevated extracellular protease activity (2.1-fold increase, *p* < 0.01). This proteolytic enhancement accelerated casein hydrolysis into bioactive peptides and free amino acids while maintaining gel structure integrity. Concurrently, quantitative EPS analysis confirmed intensified carbohydrate metabolism, suggesting BSP-mediated optimization of carbon flux partitioning between energy production and structural polymer synthesis. Notably, the intervention maintained critical quality parameters within industrial standards despite microbial hyperproliferation. Acid production kinetics remained within optimal ranges, while water-holding capacity and viscosity showed better effects compared to controls. These findings suggest that BSP-induced metabolic reprogramming enhances microbial efficiency rather than simply increasing biomass, representing a novel approach to improve fermentation efficacy without compromising product quality. The dual enhancement of proteolytic activity and EPS production positions BSP supplementation as a promising strategy for developing functionally enhanced dairy products with potential nutritional and textural advantages.

*S. thermophilus* is known for its rapid lactose metabolism, acid production, and carbon dioxide release that stimulates the growth of LAB. However, it is sensitive to acidity and experiences significant growth restriction below a pH of 5.5 [44]. In this study, the addition of BSP maintained a stable growth rate of *S. thermophilus* even after the pH dropped to 5.5, likely due to increased EPS production. EPS, produced by specific LAB species including *Streptococcus* and *Lactobacillus*, can covalently bind to the cell surface, forming capsules that enhance bacterial resilience to harsh conditions in yogurt [45,46]. Additionally, EPS contributes to the texture, firmness, and viscosity of yogurt and improves its water-holding capacity [47,48,49,50].

The addition of BSP significantly altered the metabolic profile of yogurt, leading to the accumulation of various compounds, including alcohols, aldehydes, ketones, carbonyl compounds, organic acids, and others. These metabolites are primarily derived from the lipolysis of milk fat and the microbial transformation of lactose and citric acid. Short-chain fatty acids, known for their contribution to yogurt’s characteristic flavor, were also elevated. The nutritional profile of the yogurt was enhanced by the upregulation of L-ornithine, betaine, and 1,3,5-trihydroxybenzene. Metabolic pathway analysis revealed significant upregulation of amino acid metabolism, including alanine, aspartate, and glutamate metabolism, as well as valine, leucine, and isoleucine biosynthesis. The increased levels of amino acids in the yogurt can be attributed to the enhanced activity of these metabolic pathways.

The observed metabolic pathway alterations reveal a potential sophisticated microbial adaptation strategy orchestrated by BSP supplementation (Figure 6). Among the enhanced reactions, arginine decarboxylation could increase the growth of lactobacilli [51], catalyzing the conversion of arginine to ornithine with concomitant NH_3_ release. This biochemical process serves intracellular pH homeostasis through the proton neutralization that is critical for protecting the cells against acid damage [52]. Concurrently, hypoxanthine oxidation consumes dissolved oxygen, creating an anaerobic niche essential for LAB growth, while suppressed xanthine oxidation reduces H_2_O_2_ and urate production, thereby reducing their toxic effects on LAB. The coordinated enhancement of carbamoyl-phosphate synthesis and thymidine phosphorolysis establishes a pyrimidine salvage network. The ATP-dependent conversion of L-glutamine to carbamoyl-phosphate is the rate-limiting step in de novo pyrimidine synthesis. Concurrent thymidine phosphorolysis enables nucleoside recycling from thymidine, conserving energy expenditure. This metabolic economy facilitates rapid DNA repair and helps *Mycoplasma hominis* to survive under stress [53]. The attenuation of *N*-formimino-L-glutamate deamination suppression suggests strategic redirection of one-carbon metabolism, preserving formiminotetrahydrofolate for purine biosynthesis rather than histidine catabolism [54]. These adjustments may synergistically enhance the stress resistance of cells. Although these metabolic adaptations correlate with improved bacterial viability and yogurt quality parameters, caution must be exercised in interpreting these omics-derived findings. The current dataset might contain some detected compounds that show significant matrix interference from dairy components. Furthermore, the observed pathway alterations require functional validation to distinguish causal relationships from correlative phenomena. These considerations will form the basis of our subsequent investigation to establish direct genotype–phenotype linkages in this adaptive response system.

During cold storage, the pH and cell proportions in yogurt decreased slowly, likely due to the continued activity of *L. bulgaricus* under low-temperature conditions. BSP supplementation further enhanced the viability of yogurt during storage at 4 °C. The increased EPS production may have contributed to the improved survival of bacteria during refrigeration. Probiotic yogurts offer potential health benefits, but the viability of probiotic cultures during production and storage is a significant challenge [55]. *S. thermophilus* and *L. bulgaricus* are not inherently resistant to the harsh conditions of the gastrointestinal tract (GIT) [56]. To achieve therapeutic effects, probiotic yogurts must maintain sufficient viable cell counts throughout their shelf life. A minimum of 10^6^ CFU/g of probiotics is recommended, translating to a daily intake of a 100 g serving. However, considering the loss of cells during transit through the GIT, higher initial counts are often necessary. For instance, a minimum of 10^7^ CFU/mL is generally recommended to ensure adequate probiotic delivery to the target site [57]. Several factors can influence the viability of probiotics in yogurt, including strain variation, acid accumulation, interactions with starter cultures, oxygen and hydrogen peroxide levels, and storage conditions [58,59]. In this study, *S. thermophilus* exhibited higher cold resistance than *L. bulgaricus*, aligning with previous findings [60]. Notably, BSP significantly increased the survival rate and final viability counts of both *S. thermophilus* and *L. bulgaricus* after 7 days of storage, exceeding 10^8^ CFU/g and surpassing the standards set by the US, Japan, and China. While *S. thermophilus* and *L. bulgaricus,* which were classified as starter cultures rather than probiotics by major regulatory bodies, demonstrated enhanced viability through BSP supplementation, this technological improvement does not confer probiotic status or associated health claims under current international frameworks. Nevertheless, optimizing post-fermentation bacterial counts remains operationally critical, as elevated initial viability directly correlates with sustained microbial populations during commercial shelf-life, meeting or exceeding food safety standards across key markets.

Prebiotics have been shown to protect probiotic cells from acidic and harsh environmental conditions during production and storage [61]. While microencapsulation can also improve probiotic viability and stability in yogurt and simulated GIT conditions [62], BSP offers a more accessible and practical approach. These results suggest that BSP could serve as a valuable novel postbiotic for maintaining probiotic viability in yogurt. However, this study’s practical relevance to the food industry would benefit from a clearer cost–benefit analysis and pilot-scale production data to assess the commercial feasibility of BSP supplementation.

## 5. Conclusions

This study demonstrates that the addition of BSP to yogurt enhances the fermentation process by stimulating the growth and activity of lactic acid bacteria. During refrigerated storage, BSP supplementation effectively inhibited post-acidification and prolonged the viability of the starter LAB 10-fold. Untargeted metabolomic analysis revealed a significant upregulation of various flavor compounds, including alcohols, aldehydes, ketones, and organic acids, in BSP-supplemented yogurt. Mechanistic profiling revealed five synergistic adaptation pathways: pH homeostasis via NH_3_-mediated proton neutralization and oxygen scavenging, coupled with reduced urate and H_2_O_2_ toxicity, while establishing a pyrimidine salvage network and redirecting one-carbon metabolism. These findings suggest that BSP promotes the metabolic activity of LAB, leading to the production of a diverse range of flavor compounds. From an industrial perspective, BSP’s cost profile demonstrates scalability potential. This multi-targeted enhancement of microbial viability, acidification kinetics, and organoleptic properties positions BSP as both a functional stabilizer and flavor modulator, offering dairy manufacturers a dual-purpose natural additive that addresses the key production challenges of microbial die-off during cold chain distribution.

## Figures and Tables

**Figure 1 foods-14-01806-f001:**
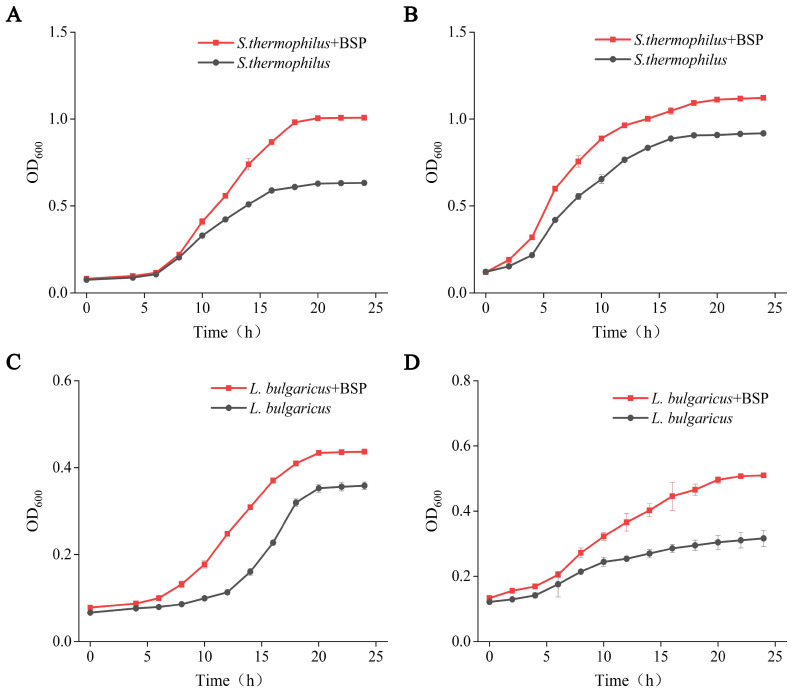
Effect of postbiotics derived from *B. subtilis* GDAAS-A32 (BSP) on the growth of *Streptococcus thermophilus* and *Lactobacillus delbrueckii subsp. bulgaricus*. (**A**) Effect of BSP on *S. thermophilus* under aerobic conditions; (**B**) effect of BSP on *S. thermophilus* under anaerobic conditions; (**C**) effect of BSP on *L. bulgaricus* under aerobic conditions; (**D**) effect of BSP on *L. bulgaricus* under anaerobic conditions.

**Figure 2 foods-14-01806-f002:**
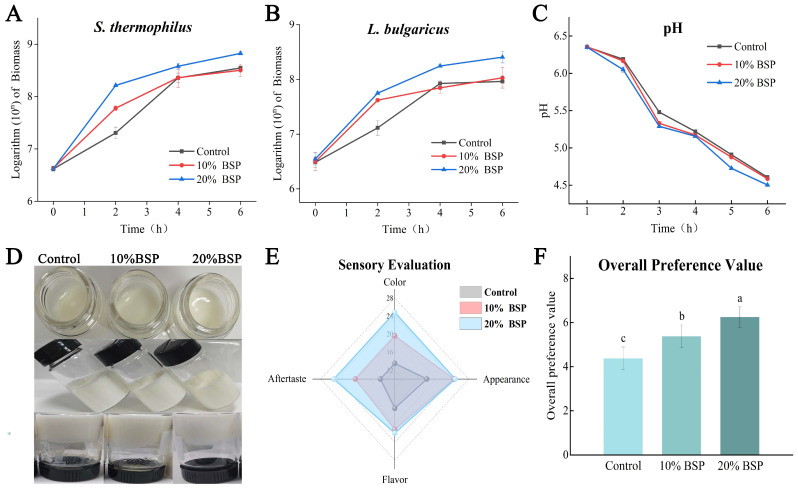
Indicators of yogurt during fermentation and quality indicators of the yogurt samples at the end of fermentation. (**A**) The biomass of *Streptococcus thermophilus*; (**B**) the biomass of *Lactobacillus delbrueckii subsp. bulgaricus*; (**C**) pH changes. (**D**) Yogurt appearance; (**E**) yogurt sensory evaluation; (**F**) yogurt overall preference value, different letters denote significant differences (*p* < 0.05).

**Figure 3 foods-14-01806-f003:**
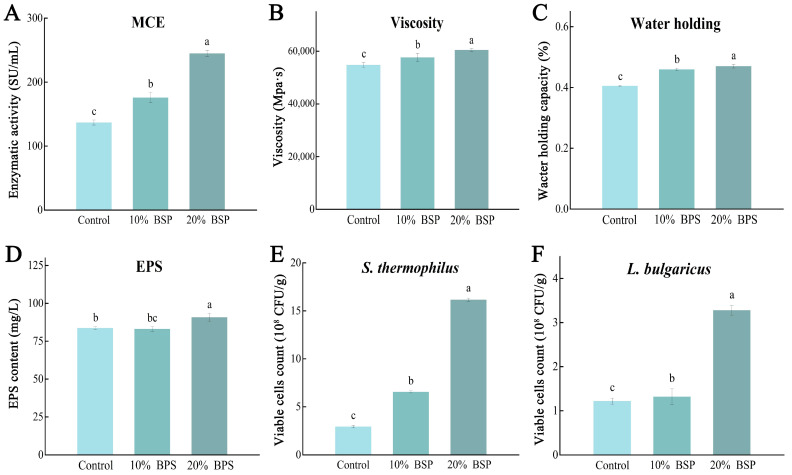
Indicators of the yogurt samples at the end of fermentation, different letters denote significant differences (*p* < 0.05). (**A**) Yogurt milk-clotting enzyme activity; (**B**) yogurt viscosity; (**C**) yogurt water-holding capacity; (**D**) yogurt EPS; (**E**) *Streptococcus thermophilus* at the end of yogurt fermentation; (**F**) *Lactobacillus delbrueckii* subsp. *bulgaricus* at the end of yogurt fermentation.

**Figure 4 foods-14-01806-f004:**
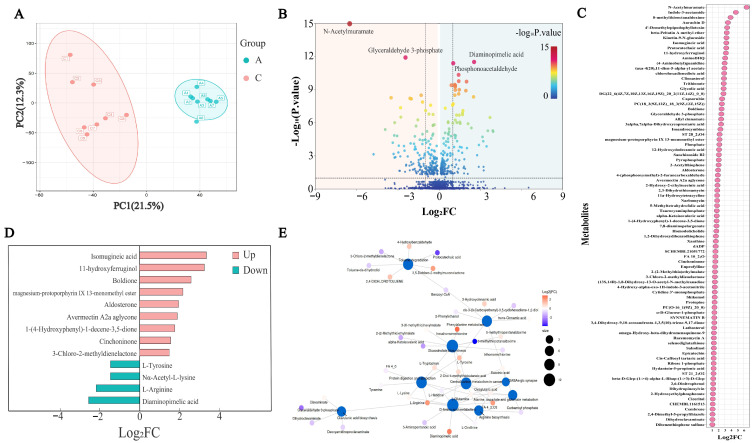
Metabolomic analysis of the yogurt samples. (**A**) Principal component analysis plot, A1–A8: control group (8 replicates); C1–C8: 20% BSP group (8 replicates); (**B**) overall volcano plot of identified metabolites; (**C**) scatter plot of metabolites with Log_2_FC greater than 1; (**D**) flavor compounds with Log_2_FC greater than 1.5; (**E**) metabolite network plot.

**Figure 5 foods-14-01806-f005:**
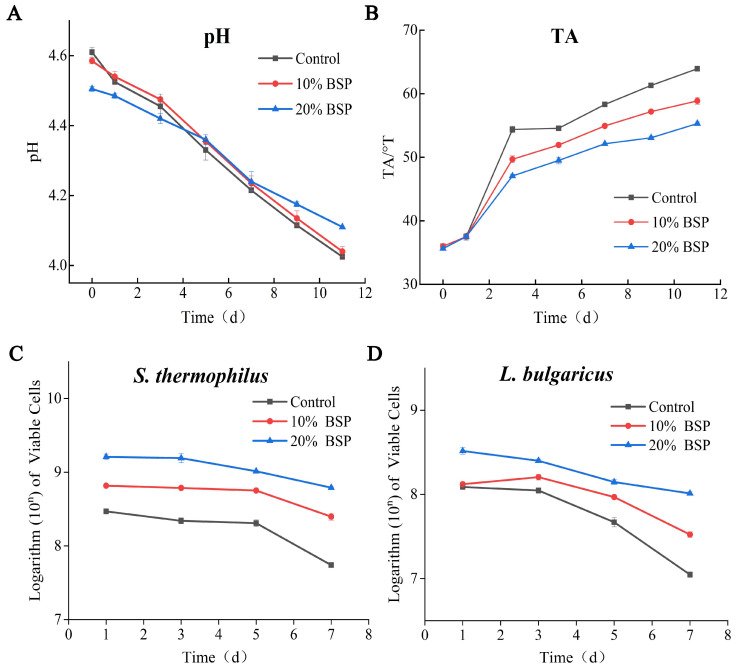
Indicators of the yogurt samples during storage. (**A**) Plot of pH changes during storage; (**B**) plot of acidity versus storage time; (**C**) viable cell counts of *Streptococcus thermophilus* during storage; (**D**) viable cell counts in *Lactobacillus delbrueckii* subsp. *bulgaricus*. during storage.

**Figure 6 foods-14-01806-f006:**
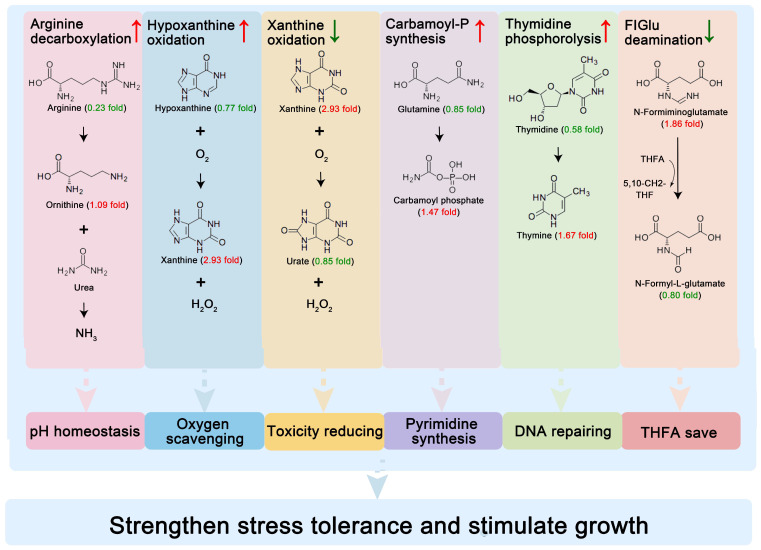
Schematic diagram of metabolic reprogramming mechanisms for enhancing lactic acid bacteria viability by *Bacillus subtilis* GDAAS-A32-derived postbiotics in yogurt.

## Data Availability

The original contributions presented in the study are included in the article, further inquiries can be directed to the corresponding author.

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
