# Peer review of "Bacillus subtilis-Derived Postbiotics as a Multifunctional Bio-Catalyst for Enhancing Lactic Acid Bacteria Viability and Yogurt Quality"

_foods, 2025, doi:10.3390/foods14101806_

Round 1
Reviewer 1 Report
Comments and Suggestions for Authors
The manuscript presents a thorough investigation on the use of Bacillus subtilis GDAAS-A32-derived postbiotics (BSP) to improve yogurt quality by enhancing lactic acid bacteria (LAB) viability. The study is relevant and timely, combining microbiological, physicochemical, sensory, and metabolomic approaches. Overall, the manuscript is well-organized and clearly written, but several critical points need to be addressed to improve its suitability.
1. Although the study demonstrates beneficial effects of BSP, the novelty is moderate. The idea of using postbiotics to enhance LAB viability has already been proposed in previous studies. The authors should give more clearly highlight in introduction how their work differs substantially from prior findings, the uniqueness of GDAAS-A32 to other postbiotics.
2. Add also the rationale for applying GDAAS-A32 in yoghurt production as an hypothesis for the aim of study.
3. Clarify whether the BSP preparation is consistent with ISAPP's definition of postbiotics. Moreover, authors should also describe more on the composition and treatment of BSP preparation. Is it prepared in sterile condition?
4. Some figures are crowded, and axis labels are small (especially in Fig 2). Please give better resolution of the Figures.
5. The results section refers to yogurt stored for 28 days and mentions significant differences in viable counts, but the methods section doesn’t describe any 28-day storage protocol. No data for extended shelf life (14, 21, or 28 days) although mentioned in methods. No figures or tables show the 28-day data. It’s unclear how the authors collected or analyzed the 28-day data if the trial was "excluded." Please clarify the methods and include corresponding figures or tables. If not, this part should be revised or removed for consistency. IF only 7 days, no need to mention 28 days experiments.
6. The authors suggest that certain pathways—like pyrimidine salvage or one-carbon metabolism—are actively triggered by BSP, but these conclusions are drawn just from metabolite level changes. Without any follow-up validation (like gene expression or enzyme activity), these claims feel more speculative than proven. It would be better to present these ideas more cautiously and make it clear that further evidence is needed to confirm the mechanisms.
7. Please check the scientific name format for microbial name, it must be written in Italic, this typo was found throughout the article
Author Response
- Thank you for the suggestion. Previously, we isolated a B. subtilis strain, GDAAS-A32, from poultry intestine, which demonstrated a strong capability to promote the growth of various LAB under both aerobic and anaerobic conditions in vitro and in vivo. The intestinal origin of GDAAS-A32 confers ecological compatibility with LAB consortia, suggesting evolutionary adaptation to microbial cross-interaction mechanisms within nutrient-rich gastrointestinal niches—a trait hypothesized to enhance its functional synergy with dairy fermentation systems.We have added this description in the introduction.
- We have added this description in the introduction.
- To comply with the ISAPP criteria defining non-viable microbial preparations, the supernatant was sterilized through 0.22-μm polyethersulfone membrane filtration to generate the BSP. Sterility verification was rigorously performed by inoculating LB with filtered BSP and monitoring microbial growth through 72-hour turbidimetric analysis at 37°C with continuous agitation (150 rpm), confirming complete absence of culturable microorganisms.We have added this description in section 2.2.
- Thank you for the suggestion. We have revised the Fig. 2, 3, and 4.
- The initial experimental protocol specified a 28-day observation period. Nevertheless, upon examination of the subsequent time point following 7 days, significant sample deterioration was detected (characterized by visible microbial growth and smell alteration), presumably resulting from exogenous microbial contamination due to cumulative exposure effects during sequential sampling operations. We have comprehensively modified and expanded the corresponding results section, please see section 3.5.
- We agree with this opinion. Although these metabolic adaptations correlate with improved bacterial viability and yogurt quality parameters, caution must be exercised in interpreting these omics-derived findings. The current dataset contains might contain some of detected compounds show significant matrix interference from dairy components. Furthermore, the observed pathway alterations require functional validation to distinguish causal relationships from correlative phenomena. These considerations will form the basis of our subsequent investigation to establish direct genotype-phenotype linkages in this adaptive response system.We have mentioned this description in the discussion.
- Sorry for these mistakes. The current version's irregularities are attributable to the file conversion process when implementing the journal's style template. A thorough proofreading and corresponding corrections have been made to the complete text.
Reviewer 2 Report
Comments and Suggestions for Authors
Title: “Bacillus subtilis-derived postbiotics as a multifunctional bio catalyst for enhancing lactic acid bacteria viability and yogurt quality”
by Wu et al., Foods
General comments: In the present work, the authors aim to demonstrate the improved viability and functionality of yogurt enriched with Bacillus subtilis. Authors investigate the yogurt fermented with and without the addition of B. subtilis from a microbiological, physicochemical and sensory point of view and explore its profile of emitted volatile organic compounds. The work is written in clear and fluent English. The methodology is well described although some references are missing. The results are well explained but the data should be shown more clearly and legibly, however, they are well discussed. Overall, the work needs major revisions. Following are my suggestions.
Specific comments:
- Please check in the text that the name of the species is in italics;
- In the introduction, the data should be corrected in the sentence “Clinical trials have demonstrated that ≥109 CFU/dose of viable LAB…”.
- In the introduction, authors might emphasise their work a bit, since it falls into the current research trend, referring in particular to already existing studies in the literature on the functional improvement of yogurt such as https://doi.org/10.1016/j.jafr.2024.101153, https://doi.org/10.1016/j.nfs.2023.100143;
- In section 2.1, include the storage conditions for the strains;
- In section 2.2, why were the strains transferred in m9 medium?
- In section 2.2, it is unclear what the authors do after incubation at 37°C, please explain better;
- In section 2.3, why were the strains transferred in modified MC medium? Insert reference;
- Insert an experimental plan in graphic form illustrating the organization of work and a flow chart followed for yogurt in section 2.4;
- In section 2.4, pasteurization at 75°C for 30 minutes appears excessive to me, was it done following any methodology seen? If yes insert reference. Moreover, why were the two strains included in this ratio? Insert reference.
- In section 2.5, strains should be incubated theoretically under anaerobic conditions.
- In section 2.5, include a reference to the PCR reaction procedure.
- In section 3.2, , the data should be corrected in the sentence “By the end of fermentation, the 20% BSP group exhibited a significant increase in S. thermophilus of 6.71×108 CFU/g..”;
- Improve the quality of the figures and put them bigger;
- In the text several times the exponent preceding CFU is not superscripted, correct it.
- Application of a starter involves screening of the whole genome of the strain to see the presence of virulence or antibiotic resistance genes, was the strain you employed previously screened for this?
- Check the formatting of the bibliography according to the guidelines of the journal.
Author Response
- Sorry for these mistakes. The current version's irregularities are attributable to the file conversion process when implementing the journal's style template. A thorough proofreading and corresponding corrections have been made to the complete text.
- A thorough proofreading and corresponding corrections for CFU data have been made to the complete text.
- Thank you for the suggestion.We have added the description of the functional improvement of yogurt and the corresponding references in the introduction.
- Thank you for the suggestion.We have added the storage conditions for the strains in section 2.1.
- The M9 minimal medium was employed to eliminate potential interference from the rich nutrient components present in LB medium. We have added this description in section 2.2.
- Sorry for this confusing sentence.We have revised this description in section 2.2.
- Preliminary assessments indicated not particularly obvious growth-promoting effects of BSP on LAB in nutrient-replete MRS medium. This attenuated response was attributed to nutritional saturation inherent to the MRS formulation, which likely obscured potential stimulatory interactions through comprehensive metabolic provisioning. To isolate BSP-mediated effects, we transitioned to a modified MC medium (5 g/L tryptone, 3 g/L beef extract, 3 g/L yeast extract, 20 g/L glucose) that supports controlled microbial proliferation. This formulation provides suboptimal nutrient availability while maintaining biomass yield stability, thereby enabling detection of growth modulation.We have added this description in section 2.3.
- Thank you for the suggestion.We have added a Fig.S1 for BSP preparation and yogurt production workflow in section 2.4, please see supplementary materials.
- Sorry for this mistake. This was a typographical error; the correct value should read 65°C. We have revised this value in section 2.4. The ratio between the two bacterial strains was determined based on established protocols in the cited literature [23]. We have inserted this reference in section 2.4.
- Preliminary comparative enumeration under both anaerobic and aerobic conditions revealed no statistically significant differences between the two approaches. Therefore, for practical considerations, aerobic plating was adopted for subsequent experiments. The validity of this method was further confirmed by strong correlation with qPCR quantification, ensuring the reliability of our enumeration data.
- The PCR amplification protocol was performed strictly according to the manufacturer's instructions. So no references were inserted here.
- A thorough proofreading and corresponding corrections for CFU data have been made to the complete text.
- Thank you for the suggestion. We have revised the Figures.
- A thorough proofreading and corresponding corrections for CFU data have been made to the complete text.
- In a previous work, B. subtilisGDAAS-A32 was isolated from poultry intestinal microbiota, demonstrating exceptional probiotic potential through its capacity to enhance LAB proliferation [22]. While chromosomal architectures showed conserved synteny (99.98% similarity), GDAAS-A32 harbored anendogenous plasmid absent in the reference strain 168. Comprehensive biosecurity assessment via pathogenicity island prediction (PIPS), virulence factor database (VFDB) and comprehensive antibiotic resistance database (CARD) screening confirmed the absence of virulence factors and antibiotic resistance genes in the plasmid. We have added this description in the discussion.
- Sorry for these mistakes. We have revised the format of the bibliography.
Round 2
Reviewer 2 Report
Comments and Suggestions for Authors
Authors have adequately answered the given suggestions. The work is now ready for publication.